# Replication Compartments of DNA Viruses in the Nucleus: Location, Location, Location

**DOI:** 10.3390/v12020151

**Published:** 2020-01-29

**Authors:** Matthew Charman, Matthew D. Weitzman

**Affiliations:** 1Division of Protective Immunity and Division of Cancer Pathobiology, Children’s Hospital of Philadelphia, Philadelphia, PA 19104, USA; charmanm@email.chop.edu; 2Department of Pathology and Laboratory Medicine, University of Pennsylvania Perelman School of Medicine, Philadelphia, PA 19104, USA

**Keywords:** DNA virus, replication compartment, replication center, nucleus, DNA replication, biomolecular condensate, phase separation

## Abstract

DNA viruses that replicate in the nucleus encompass a range of ubiquitous and clinically important viruses, from acute pathogens to persistent tumor viruses. These viruses must co-opt nuclear processes for the benefit of the virus, whilst evading host processes that would otherwise attenuate viral replication. Accordingly, DNA viruses induce the formation of membraneless assemblies termed viral replication compartments (VRCs). These compartments facilitate the spatial organization of viral processes and regulate virus–host interactions. Here, we review advances in our understanding of VRCs. We cover their initiation and formation, their function as the sites of viral processes, and aspects of their composition and organization. In doing so, we highlight ongoing and emerging areas of research highly pertinent to our understanding of nuclear-replicating DNA viruses.

## 1. Introduction

The DNA viruses encompass a range of ubiquitous and clinically important viruses, from acute pathogens to persistent tumor viruses. Typically, their replication occurs within the nucleus of an infected cell, utilizing host nuclear machinery and co-opting nuclear processes for the benefit of viral replication. At the same time, these viruses must evade and antagonize antiviral and homeostatic processes that would otherwise attenuate or restrict viral replication. Accordingly, DNA viruses induce the formation of nuclear assemblies termed viral replication centers or viral replication compartments (VRCs). These VRCs provide a dedicated environment in which viral processes can be organized and regulated, where factors required for viral processes are enriched, and factors that would attenuate these processes are excluded. Unlike some cytoplasmic-replicating viruses that can manipulate cellular membranes to form viral compartments, nuclear-replicating DNA viruses induce the formation of membraneless assemblies that resemble cellular membraneless nuclear sub-domains.

Here, we review advances in our understanding of VRCs of nuclear-replicating DNA viruses, covering their initiation and formation, their function as sites of viral processes, and aspects of their composition and organization. We focus on those DNA viruses that have been most studied, including members of the *Herpesviridae*, *Adenoviridae*, *Parvoviridae*, *Polyomaviridae*, and *Papillomviridae*. We provide an overview of VRCs, discussing features that DNA viruses share in common, as well as fundamental differences. In doing so, we highlight exciting emerging areas of research.

## 2. The Initiation and Formation of Replication Compartments

DNA viruses that replicate in the nucleus exhibit a diverse array of strategies to complete their replication cycle and ultimately produce progeny virions. Despite their many differences, the strategies employed by DNA viruses to initiate productive infection and establish VRCs share many features in common. Indeed, it is likely that all DNA viruses that replicate in the nucleus form VRCs. Following penetration of the host cell and the transport of viral capsids/cores, viral DNA genomes enter the nucleus, where they begin a program of temporally regulated gene expression that initiates productive infection [1,2]. Early gene products include viral proteins required to manipulate the host–cell environment and replicate the viral genome. Viral replication proteins interact with the viral genome and initiate genome replication leading to VRC formation, which is often in concert with certain co-opted host proteins. However, these viruses must overcome several challenges to form VRCs successfully. Firstly, incoming genomes must avoid cellular intrinsic antiviral defenses and homeostatic regulatory pathways such as elements of the DNA damage response (DDR) that respond to the presence of foreign DNA and act to suppress viral gene expression [3,4,5,6,7,8,9,10]. Secondly, VRCs typically form at specific sites within the nucleus, suggesting that viral genomes need to be targeted to these sites in order to initiate VRC formation [6,11,12]. Furthermore, it has been hypothesized that the formation and growth of VRCs may require manipulation of the nuclear environment and the re-organization of host chromatin to overcome the spatial restraints of the nucleus [11,12,13,14,15]. In this section, we review key features of VRC initiation and highlight some major challenges to VRC formation. 

## 3. Interplay between Viral Replication Compartment Formation and Promyelocytic Leukemia Protein Nuclear Bodies

A common theme amongst many nuclear-replicating DNA viruses is initiation of VRCs at sites in close proximity to promyelocytic leukemia protein (PML) nuclear bodies (NBs). Since the discovery that the polyomavirus simian virus 40 (SV40) and human adenovirus (HAdV) initiated VRC formation at the periphery of PML NBs, a wealth of evidence now suggests that for many DNA viruses including members of the *Herpesviridae*, *Adenoviridae*, *Parvoviridae*, *Polyomaviridae* and *Papillomviridae*, VRC formation and many subsequent viral processes take place at these sites. Much of this evidence comes from the localization of viral proteins, viral nucleic acid, and even incoming viral genomes [6,11,14,16,17,18]. One notable exception may be the autonomous parvoviruses. In particular, H-1 parvovirus forms VRCs known as autonomous parvovirus-associated replication (APAR) bodies that are distinct from PML NBs [19,20,21]. However, it should also be noted that PML co-localizes with the APAR bodies of minute virus of mice (MVM), suggesting that PML may be recruited to APAR bodies once they are established [19]. 

PML NBs function in a number of important cellular processes including DNA repair, transcription regulation, cell senescence, apoptosis, the interferon response, and intrinsic antiviral immunity [3,22,23,24]. PML NBs have also been proposed as major sites of post-translational modification of proteins by the small ubiquitin-like modification (SUMO), with the SUMO modification of PML NB proteins also playing a role in intrinsic antiviral responses to incoming viral genomes [25,26,27,28]. Interestingly, whilst DNA viruses typically initiate VRC formation at or in close proximity to PML NBs, many PML NB proteins contribute to the intrinsic antiviral restriction of DNA viruses if their function is not antagonized by the virus. These proteins include PML itself, the transcription corepressor and histone chaperone death domain associated protein 6 (DAXX), transcriptional regulator ATRX, transcription co-regulator Sp100, and E3 SUMO-protein ligase PIAS1 [3,24,28,29,30,31,32,33,34]. This has led many to questions regarding whether the spatial relationship between VRCs and PML NBs is the result of viral strategies to target these nuclear domains or antiviral strategies that target viral genomes [3,6,24]. However, it is clear that many DNA viruses disrupt or manipulate PML NBs to promote successful infection.

The interplay between incoming viral genomes and PML NBs has been particularly well studied in the context of herpes simplex virus 1 (HSV-1) infection. PML and other PML NB proteins re-localize to sites associated with incoming HSV-1 genomes, suggesting that these components are recruited to incoming genomes [27,29,35]. Consistent with these findings, the labeling of viral genomes with 5-Ethynyl-2’-deoxyuridine (EdU) combined with click chemistry demonstrates that PML can surround and encapsulate incoming viral genomes [29]. The HSV-1 immediate early gene product infected-cell protein 0 (ICP0) is a viral ubiquitin ligase that facilitates viral gene expression and reactivation from latency [35,36]. ICP0 transiently localizes to PML NBs, mediating the proteasome-dependent degradation of PML and Sp100 and the disruption of PML NB through the E3 ubiquitin ligase activity of its really interesting new gene (RING) domain [37,38,39]. In the absence of ICP0, PML and other PML NB proteins act to suppress HSV-1 gene expression and restrict infection, highlighting the intrinsic antiviral function of PML NBs and the requirement of HSV-1 to antagonize this aspect of PML NB function [3,24,27,28,29,31,40]. The necessity to antagonize the PML NB function appears to be shared by most if not all herpesviruses. Homologues of ICP0 in different alpha-herpesviruses also appear to target PML NB proteins [39,40]. For example, the varicella zoster virus (VZV) Orf61 protein disrupts PML NBs to antagonize sequestration of viral nucleocapsids within PML ‘cages’ [41,42]. Furthermore, members of the beta-herpesviruses such as cytomegalovirus (CMV) as well as members of the gamma-herpesviruses such as Kaposi’s sarcoma-associated herpesvirus (KSHV) and Epstein–Barr virus (EBV) also utilize immediate–early or early proteins that disrupt PML NB function, either through the disruption of PML NBs, or through the mislocalization or degradation of PML NB proteins [3,30,31,39].

The interplay between PML NBs and initiation of VRCs is also relatively well characterized in the context of HAdV infection. Drawing parallels with herpesviruses, HAdV early proteins antagonize the repression of viral gene expression by PML NB proteins. The E1B-55K and E4orf6 proteins facilitate the degradation of DAXX, ATRX, and Bloom syndrome protein BLM [43,44,45,46,47], while E4orf3 is responsible for a drastic redistribution of PML from spherical PML NBs into PML ‘tracks’ [17,48,49]. Interestingly, some PML NB proteins including NDP55, specific isoforms of Sp100, and SUMO2/3 are re-localized to VRCs during adenovirus infection [17,50,51]. Sequestering proteins within VRCs may act to antagonize their function, or alternatively, these proteins may be utilized for viral replication [11,17,50,51]. Polyomaviruses seemingly demonstrate an even more complex relationship with PML NBs. On the one hand, SV40 and JC virus (JCV) do not appear to disrupt PML NBs nor redistribute its key components [16,52]. The SV40 large T antigen (TAg) localizes to PML NBs during infection, or when ectopically expressed, and the expression of TAg alone is sufficient to localize reporter constructs containing the SV40 origin of replication to PML NBs [53]. Thus, it first appears that the targeting of SV40 to PML NBs is the result of a viral strategy to exploit these sites. However, somewhat paradoxically, reducing PML levels has been shown to promote SV40 or JCV infection, respectively [52,54]. In addition, VRC formation and infectious virion production are not attenuated when PML negative mouse embryonic fibroblasts or mice are infected with murine polyomavirus (MPyV), suggesting that productive infection is not dependent on PML NBs [55]. In contrast to SV40 and JCV, BK virus (BKV) does appear to counteract PML NB function by manipulating them. BKV infection results in a decrease in the number of PML NBs per nucleus with a corresponding increase in PML NB size [13]. PML NB composition is also changed, with a redistribution of Sp100 and DAXX. Although the knockdown of PML does not significantly impact BK virus infection, BKV is able to complement the growth of an ICP0-null HSV-1 mutant that is incapable of antagonizing PML NBs, suggesting that BKV may manipulate PML NBs to antagonize antiviral functions [13]. In contrast to many other DNA viruses, papillomaviruses, specifically bovine papillomavirus 1 (BPV1) and human papillomavirus (HPV) types 16 and 18, require PML for efficient viral transcription [56,57,58]. PML, SUMO1, and Sp100 are recruited to incoming HPV genomes [59]. However, although PML functions in a pro-viral role, Sp100 restricts HPV 18 transcription and replication [58,60]. Thus, papillomaviruses may both utilize and antagonize PML components to replicate.

Considering these findings with different DNA viruses, it seems likely that there are both benefits and drawbacks associated with VRC formation at the sites of PML NBs and that DNA viruses have developed different strategies to manipulate PML NBs accordingly. 

## 4. What Dictates the Number of Replication Compartments that Form in Each Infected Nucleus?

Under experimental conditions, DNA viruses typically initiate multiple VRCs with numbers anywhere between 1 and 20 per nucleus depending on the virus in question, experimental conditions such as multiplicity of infection (MOI), and the cell type infected [16,61,62,63,64,65,66,67]. In the case of HAdV, the number of initiated VRCs correlates with MOI, up to a point of saturation where the addition of more virus does not result in an increase in VRC number [68]. This suggests that the number of VRCs formed is dependent on the number of incoming viral genomes, with each VRC likely initiated by a single genome. Additionally, this suggests that the cell is only capable of supporting a limited number of VRCs. The visualization of incoming HSV-1 genomes using 5-Ethynyl-2’-deoxycytidine (EdC) labeling and click chemistry demonstrates that incoming viral genomes are associated with spatially distinct replication center foci at early times post-infection [63,64]. This confirms that in the context of HSV-1 infection, the majority of VRCs likely initiate from a single genome. During co-infection with different pseudorabies viruses (PRV), PRV and HSV-1 in combination, or different HSV-1 recombinants, each virus is typically replicated within distinct territories [61,62,69]. Interestingly, this not only suggests that each VRC is initiated by a viral genome, but that each VRC contains only replicated progeny derived from the initial founder virus. The number of PRV genomes that initiate replication in each cell is also MOI dependent, with an upper limit of approximately eight genomes, correlating with the number of VRCs typically initiated per nucleus [61,65]. Together, these observations support a general model in which VRCs of DNA viruses are initiated from single viral genomes that have successfully evaded cellular responses and undergone replication.

Although it is not entirely clear what dictates the upper limit of how many VRCs can form per nucleus, limitations on the availability of cellular factors required to initiate viral genome replication and the number of available sites viable for VRC initiation have been proposed as contributing factors [11,25,69]. Interestingly, both the number and size of HSV-1 VRCs correlates with nucleus size, suggesting that there may be spatial restraints to how many VRCs can form in an infected nucleus [62]. Indeed, it is possible that the upper limit of VRC number is restricted by availability of sites viable for the initiation and growth of VRCs. Interestingly, cellular chromatin is excluded from the VRCs of many DNA viruses, including HSV-1, CMV, and HAdV [11,12,14,15]. This raises the possibility that cellular chromatin must be manipulated to make space for VRC growth. Many DNA viruses hijack cellular chromatin modifiers to regulate their own viral chromatin, as well as cellular chromatin [70,71,72]. Furthermore, many DNA viruses activate the DDR, leading to the phosphorylation of histone γH2AX, which is a key mediator of cellular chromatin remodeling and condensation during DNA repair [8,9,10]. In addition, some DNA viruses such as HPV and HAdV encode viral proteins that can manipulate cellular chromatin [70,73,74,75]. For example, expression of the HAdV core protein VII alone is sufficient to manipulate cellular chromatin [74]. Given that DNA viruses can manipulate cellular chromatin, it is tempting to speculate that the exclusion of chromatin from VRCs may be the result of viral strategies to promote VRC formation.

## 5. Viral Processes that Initiate Replication Compartment Formation

Early observations that viral DNA and viral proteins localized to dedicated sites of VRC initiation raised the idea that many or even all viral processes might be compartmentalized within these sites. However, considering findings with HSV-1 and HAdV, it seems likely that the transcription of incoming viral genomes can begin prior to their localization at VRC initiation sites. For example, incoming genomes of WT HSV-1 transiently co-localize with PML NBs prior to the disruption of PML NBs by ICP0 [25,29,40]. The disruption of PML NBs requires ICP0 that is generated de novo [36,37,40,76]. Thus, some viral immediate early gene expression must occur prior to or irrespective of the entrapment of viral genomes within PML NBs. It has also been suggested that incoming viral genomes are targeted to the nuclear lamina prior to or at the stage of immediate early gene expression and that this targeting is required for efficient early gene expression [77]. Investigations into the localization of incoming HAdV genomes indicate that incoming genomes are not associated with PML NBs immediately following nuclear entry. Instead, the subsequent co-localization of VRCs with PML NB proteins results from viral single-stranded DNA (ssDNA) binding protein (DBP) targeting and recruiting PML NB components in a DNA replication-dependent manner [3,78]. Thus, it seems probable that viral gene expression occurs prior to genomes localizing to VRC initiation sites. 

Viral gene expression is essential for VRC formation. The recruitment of viral and cellular proteins required to initiate viral DNA replication leads to the further recruitment of replication factors and further genome amplification. Thus, the initiation and growth of VRCs is intimately linked with and largely synonymous with DNA replication. Accordingly, the events that dictate VRC formation are perhaps best described by biochemical and molecular studies that have defined mechanisms of viral genome replication in the case of each respective virus. These mechanisms are largely outside of the scope of this review but have been extensively reviewed elsewhere [79,80,81,82,83]. However, one particularly illustrative example of VRC initiation is the stepwise disruption of PML NBs and recruitment of replication factors to VRCs during HSV-1 infection. Burkham et al. describe four stages of initiation and formation of HSV-1 VRCs [84,85,86]. Stage I: PML NBs are intact. Stage II: PML NBs are disrupted, coinciding with detection of the viral ssDNA binding protein infected-cell protein 8 (ICP8) diffusing throughout the nucleus, which is indicative of viral protein synthesis. Stage III: ICP8 containing foci form. Stage IV: Fully fledged VRCs form containing ICP8 and the other viral proteins required for viral genome replication. Stage III is dependent on primase helicase complex (UL5, UL8, and UL52), and origin binding protein (UL9), since cells infected with mutants lacking these proteins fail to progress to Stage III [84,85,86]. Progression to Stage IV requires the binding of HSV-1 polymerase and UL42 accessory protein to the nucleoprotein complex [84,85,86]. These findings illustrate the recruitment of replication factors to initiating viral genomes at pre-replication foci and the subsequent amplification of viral genomes that gives rise to full VRCs. However, perhaps the most interesting question remains: How are the factors required for DNA replication driven to VRC initiation sites?

## 6. Biophysical Processes in Replication Compartment Formation

An exciting emerging area of research is the study of liquid–liquid phase separation (LLPS) in the organization of intracellular compartments. Many membraneless organelles including nuclear assemblies such as PML NBs, Cajal bodies, and nuclear speckles represent biomolecular condensates (BMCs) formed through LLPS. Their formation is thought to be driven by transient protein–protein or protein–nucleic acid interactions, and it can be facilitated by the intrinsically disordered regions (IDRs) of proteins [87,88,89,90]. 

Recent advances suggest that LLPS may play an important role in some viral processes. For example, the cytoplasmic inclusions that harbor vesicular stomatitis virus (VSV) RNA synthesis demonstrate several liquid-like properties, including the ability to fuse together, and the rapid exchange of viral polymerase complex protein P between these inclusions and the cytoplasm [91]. In addition, influenza A virus (IAV) ribonucleoproteins form liquid inclusions at endoplasmic reticulum exit sites, suggesting that LLPS may play a role in the early stages of IAV assembly [92]. Furthermore, the ability of the Hendra virus V protein to form liquid hydrogels that may contribute to cellular toxicity was recently demonstrated [93]. However, although it is well established that nuclear-replicating DNA viruses disrupt existing nuclear biomolecular condensates such as PML NBs, Cajal bodies, and nucleoli [14,25,73,94], if and how LLPS influences the compartmentalization and organization of viral processes, including the formation of VRCs, is unknown. It has been hypothesized that the VRCs of DNA viruses might form via LLPS, as these virus-induced structures demonstrate many characteristics of BMCs [69,95]. For example, many viral proteins including VRC proteins contain either predicted or experimentally validated IDRs [96,97,98,99,100,101]. In many cases, the accumulation of these viral proteins throughout the nucleoplasm can be detected prior to the formation of VRCs [16,68,85], suggesting that their formation is concentration dependent, which is another feature of LLPS [87,88,89,90]. HSV-1, CMV, PRV, and HAdV VRCs also coalesce as infection progresses and they grow, fusing together in a liquid-like manner [14,15,61,62,65,69,95,102,103,104]. However, it was recently demonstrated that HSV-1 VRCs are not disrupted by treatment with 1,6-hexanediol, which is a feature of many other cellular BMCs. Moreover, the authors report that the recruitment of RNA polymerase II (RNAP II) to VRCs and its movement within VRCs are best explained by a model other than LLPS [105]. However, the role of LLPS in biological processes can be subtle and complex. For example, LLPS can generate liquid-like intermediates that lead to gel-like, and solid assemblies that do not themselves behave similar to a liquid [87,89]. Given that DNA virus infection presents an attractive model to study the de novo formation of membraneless nuclear assemblies, it is likely that the role of LLPS and other biophysical processes in VRC formation will remain an area of great interest.

## 7. The Composition of Replication Compartments

Understanding the composition of VRCs may inform us about both their formation and functions. Much work has been done to study the composition and architecture of cytoplasmic VRCs, including those of RNA viruses and nucleocytoplasmic large DNA viruses. That fact that cytoplasmic viral compartments exist as membrane-bound assemblies has greatly facilitated their isolation and characterization [106,107,108,109,110,111,112,113,114]. However, the VRCs of nuclear-replicating DNA viruses exist as membraneless assemblies [11,69]. Thus, their isolation presents a different and technically demanding challenge compared to the isolation of their cytoplasmic counterparts. Despite these technical difficulties, there has been some success with the isolation of HAdV VRCs. Velocity gradient centrifugation has been used to isolate cell-free fractions enriched for known VRC components. These fractions are morphologically similar to VRCs in situ and support viral DNA replication as well as the synthesis and splicing of viral RNA [115,116]. However, an in-depth analysis of the composition of these fractions was not performed as part of these studies.

Several key studies have contributed significantly to our understanding of the proteins present at VRCs using methods to isolate viral replication complexes in association with their interaction partners followed by mass spectrometry (MS) analysis of the isolated proteins. The immunoprecipitation of HSV-1 ICP8 in combination with MS identified over 50 viral and host interactors. These included host proteins with known functions in DNA replication, DNA repair, and chromatin remodeling [117]. Other approaches have utilized the isolation of proteins on nascent DNA (iPOND), specifically labeling replicating viral DNA during infection with HSV-1, HAdV, or a cytoplasmic-replicating pox virus, and isolating labeled DNA in association with proteins. Similarly to the immunoprecipitation of ICP8, these studies identified host proteins involved in DNA replication, chromatin remodeling, DNA repair, transcription, and RNA processing, as well as nucleolar proteins [18,118,119]. Furthermore, this approach has provided insight into host factors that are excluded from VRCs to prevent the restriction of viral replication [18].

Many cellular DDR proteins including DNA repair and DNA replication factors localize to VRCs where they are proposed to play roles in viral genome replication, the resolution of genome-replication intermediates, and the processing of viral genomes for packaging [8,10,11,14,18,69,120,121,122,123]. Thus, DDR proteins may be considered common components of VRCs. While some DDR pathways are antagonized by DNA viruses to prevent the DDR from attenuating infection, others are activated by viruses to promote viral processes [8,9,10,124,125]. However, exactly how cellular DNA repair and DNA replication factors are harnessed to promote viral processes is often unclear. It is also of note that many DNA viruses recruit the cellular ssDNA-binding protein complex replication protein A (RPA) to VRCs [8,10,11,14,18]. RPA plays key roles in cellular DNA replication and repair, and it likely facilitates the recruitment of DNA repair and DNA replication factors to viral genomes to promote viral processes [8,10,126]. Interestingly, the recruitment of RPA to VRCs is also true of HAdV and herpesviruses, which encode their own viral ssDNA-binding proteins [14,80]. RPA can substitute for HAdV DBP when AAV DNA replication is recapitulated in vitro [127,128], and it can promote AAV replication in infected cells [129]. This raises interesting questions as to the exact role of RPA during HAdV and herpesvirus replication.

Ultimately, many viral and cellular proteins recruited to VRCs have been identified, informing the functions of these virus-induced compartments. However, how recruited proteins function in viral processes is not always clear. Furthermore, without a global picture of VRC composition, it is likely that important aspects of VRC biology remain unknown.

## 8. Viral Processes that Take Place at Replication Compartments

Following their initiation, a wealth of different viral and cellular biomolecules localize to VRCs, providing insight into the plethora of viral and co-opted cellular processes that take place at these sites. Universally amongst nuclear-replicating DNA viruses, these VRCs are sites of viral genome replication, compartmentalizing and therefore facilitating the organization and regulation of viral DNA replication. In addition, VRCs facilitate the spatial organization of other key viral processes including viral gene expression, the assembly of viral particles, and the packaging of viral genomes. In this section, we review key evidence for the localization of these processes at or in proximity to VRCs and discuss the function of VRCs as the sites of viral processes during productive infection.

## 9. DNA Replication

Consistent with the notion that VRCs are dedicated sites of viral genome replication, viral and cellular proteins required for DNA replication are recruited to VRCs. In the case of herpesviruses including HSV-1, HCMV, KSHV, EBV, and VZV, viral proteins involved in DNA replication have been identified at VRCs. These include ssDNA-binding proteins, DNA polymerases, polymerase-associated processivity factors, and primase/helicase proteins [11,15,69,84,85,86]. Of note, the earliest identification of VRCs resulted from observation of the DNA replication-dependent localization of ICP8 to discrete nuclear sub-domains in HSV-1-infected cells [130]. In the case of HAdV, the three viral proteins that are essential for DNA replication, namely, pre-terminal protein (pTP), viral polymerase (AdPol), and DBP, as well as co-opted replication-enhancer nuclear factor 1 (NF1) localize to VRCs [16,131,132,133,134]. Furthermore, the replication proteins of AAV localize to HAdV VRCs in co-infected cells, as this virus usurps these structures to replicate its genome [121]. However, interestingly, these rep proteins localize to distinct AAV VRCs in cells co-infected with HSV-1 helper virus [135]. In the case of DNA viruses that utilize cellular DNA polymerases for their genome replication, viral proteins that facilitate replication localize to VRCs. For example, this includes the E1 helicase and E2 transcription/replication factors of papillomaviruses as well as the TAg of polyomaviruses [13,55,136,137,138,139,140].

In addition to the localization of DNA replication machinery to VRCs, viral DNA also accumulates at these compartments, as first identified during HSV-1 infection [141]. Viral DNA can be visualized using fluorescent in situ hybridization (FISH) [53,55,121,131,132] or techniques that utilize the labeling of DNA through the incorporation of the nucleotide analogues 5-bromo-2’-deoxyuridine (BrdU) or EdU during DNA replication [141,142]. Since many DNA viruses induce the shutoff of host DNA replication, BrdU or EdU are preferentially incorporated into viral DNA during infection. These techniques have been used to identify viral DNA at VRCs during infection with a range of different DNA viruses [18,55,118,119,138,142,143,144,145,146]. Using short pulses of BrdU or EdU during virus infection, it is possible to label and thus visualize replicating (nascent) DNA, thus identifying the sites of DNA replication (Figure 1). Using this technique, nascent DNA has been visualized at VRCs of several DNA viruses, including HSV-1, CMV, HAdV, BKV, and SV40 [18,104,118,119,141,146,147,148]. This has confirmed VRCs as *bona fide* sites of DNA synthesis.

## 10. Transcription

While viral early gene expression occurs prior to VRC formation, it is clear that once they have formed, VRCs facilitate the spatial organization of continued viral transcription. A number of viral proteins that facilitate viral transcription are known to localize to VRCs. For example, these include HSV-1, ICP0, infected-cell proteins 4 (ICP4), 22 (ICP22), and 27 (ICP27), and products of the CMV UL112-113 gene [11,15,64,149,150,151]. The localization of viral proteins that facilitate both DNA replication and transcription, as is the case for polyomavirus large T antigen and papillomavirus E2 and E3 proteins, further support a case for VRCs as sites of both viral DNA replication and transcription [11,55,137,138,139,140,152,153]. In the case of adenovirus, transcription of the viral genomes is believed to occur not within the center of VRCs but rather at their periphery, as evidenced by the localization of cellular factors involved in transcription and RNA biogenesis proximal to VRCs [14,132]. These include the ASF/SF2 splice enhancer, heterogeneous nuclear ribonucleoprotein A1 (hnRNP A1), and small nuclear ribonucleoproteins (snRNPs) [154,155,156,157].

DNA virus transcription is dependent on components of the cellular transcription machinery. Accordingly, RNAP II has been shown to localize to VRCs during infection with a number of DNA viruses, including HSV-1 and HCMV [11,105,158,159,160]. In addition to the visualization of viral and cellular proteins involved in transcription, it is also possible to detect viral RNA using FISH or the incorporation of nucleotide analogue similarly to BrdU and EdU labeling of DNA [53,68,132,161]. For example, short pulses of 5-Bromouridine (BrU) labeling reveal nascent RNA at the periphery of VRCs during adenovirus infection [132]. Together, this evidence demonstrates that VRCs act not only to compartmentalize viral DNA replication, but also compartmentalize or organize viral transcription. The identification of factors involved in transcriptional and RNA processing on the replicating genomes of HSV-1 and adenovirus is consistent with this notion [18,118,119]. This suggests a close spatial and functional relationship between the processes of DNA replication and transcription. 

## 11. Virion Production at Replication Compartments

The ultimate goal of productive virus infection is to produce de novo infectious virions and propagate infection. This requires the assembly of viral structural proteins into viral particles and the packaging/encapsulation of viral genomes within particles. Although a wealth of evidence supports a role for VRCs as the sites of DNA replication and transcription, direct evidence that these nuclear sub-compartments also function as sites of virion production is more limited. Perhaps some of the best evidence for VRCs as sites of virion production comes from studies of the polyomaviruses, which have identified viral capsid proteins and even accumulated viral capsids at VRCs. For example, VP1 and VP2/3 capsid proteins of BKV localize to VRCs during infection [146]. Furthermore, VP1, VP2, and VP3 capsid proteins of JCV co-localize with PML, which are the sites of VRC formation, when ectopically expressed [162]. Electron microscopy of MPyV and JCV-infected cells reveal accumulated virions and adjacent tubular structures that appear to be assembly intermediates from which assembling virions “bud”. These virions and tubular structures localized to inter-chromosomal spaces that stain positive for PML and VP1, suggesting that polyomavirus VRCs are the sites of active capsid assembly [55,163]. In the case of papillomaviruses, although the spatial organization of capsid assembly and packaging is not well understood, a number of studies have identified the localization of L1 major and L2 minor capsid protein at VRCs similarly to the localization of polyomavirus capsid proteins [137,152,164].

In the case of HAdV, the sites of particle assembly and packaging have yet to be experimentally determined. However, many viral structural proteins and packaging are excluded from VRCs, localizing instead throughout the nucleoplasm, at the periphery of the nucleus, or at distinct foci [103,145,165]. The adenovirus 52-55K protein is transiently incorporated into assembling viral particles, and it is required for either the packaging or retention of viral genomes [165,166,167,168]. A recent study has identified the viral late protein 52-55K, as well as assembled virions and assembly intermediates at the periphery of the VRC where Ad genome replication takes place, suggesting that adenovirus particles are assembled and packaged at these sites [165]. In the case of herpesviruses, in particular HSV-1 and CMV, the assembly and packaging of viral particles is proposed to occur within VRCs [11,15,76]. Several HSV-1 capsid proteins can be detected in VRCs, including VP22, VP23, and VP5 hexamers; the latter are present on the surface of assembled capsids [169,170,171,172]. Furthermore, packaging protein UL32 is required for the localization of VP5 hexamers to VRCs, inferring that UL32 is required to localize assembled capsids at VRCs so that they can be packaged [170]. However, given that direct evidence indicating where HSV-1 capsids assemble is lacking, it is unclear whether these observations represent the assembly of capsids at VRCs or their accumulation at these sites following assembly elsewhere. It is also interesting to note that HCMV UL56 protein involved in genome packaging is recruited to VRCs in a DNA replication-dependent manner, suggesting that the processes of DNA replication and packaging are linked [173]. 

In summary, evidence in support of VRCs as sites of virion assembly varies from virus to virus. Given that viral genomes required for progeny production are generated at VRCs, the production of viral progeny must inevitably be linked to VRC function to some extent. The visualization and tracking of viral genomes post-replication may yet help to define the link between replication and packaging as well as provide greater insight into where viral genome packaging takes place.

## 12. Structure of Replication Centers and Organization of Viral Processes

Detailed models describing the central role of VRCs in viral replication cycles have been illustrated for a number of DNA viruses [11,95]. However, there is much that we do not know about their structure or the spatial organization of viral processes within them. Even less is known about how the structure and organization of VRCs change as infection progresses. Adenovirus VRCs are perhaps the best characterized with regard to their spatial organization [95]. Viral ssDNA accumulated as a replication intermediate during adenovirus genome replication, and DBP are present within well-delimited, compact fibrillar structures termed single-stranded DNA accumulation sites (ssDAS). BrdU labeling of replicating DNA identified ongoing viral genome replication within the ssDAS in only a limited number of VRCs, leading to the suggestion that replication is intermittent within the ssDAS. In contrast, viral genome replication is continuous in the surrounding fibrillo-granular areas of the nucleoplasm, which is termed the peripheral replicative zone (PRZ) [131]. Consistent with this interpretation, AdPol and pTP predominantly localize to PRZs [133]. DNA replicated at the PRZ subsequently moves out to the surrounding nucleoplasm, where viral RNA as well as cellular RNA processing factors localize [132]. This suggests that the transcription of viral genomes and viral RNA processing occurs in proximity to the PRZ. Adenovirus is believed to package its genomes into pre-formed capsids, as described by a sequential model of packaging similar to HSV-1 or certain bacteriophages [76,166,168]. However, recent findings identify viral DNA, the 52-55K packaging protein, formed particles, and proposed assembly intermediates within the PRZ [165]. Thus, the authors suggest that the PRZ may actually be the site of assembly and packaging via a proposed concerted model whereby assembly and packaging occur concurrently [165]. In the case of herpesviruses, viral genome replication and ongoing viral transcription may take place at distinct sites within the VRC. A recent study using stimulated emission depleted (STED) microscopy to achieve super-resolution images of VRCs confirmed that ICP8 (genome replication) and ICP4 (transcription activation) occupy discrete domains within HSV-1 VRCs [174]. Furthermore, in the case of HCMV, replicating DNA localizes to the periphery of VRCs, while accumulated DNA localizes to the VRC interior [175]. Thus, it has been suggested that HCMV genomes are replicated at the periphery of VRCs before moving inwards to be packaged into assembled particles.

It has also become apparent that HAdV VRCs are progressively re-organized as the late phase of infection progresses. They transition from a discrete rounded morphology to a more broken and diffuse morphology and finally to ring-like assemblies [14]. The morphological changes coincide with the formation of virus-induced post-replicative (ViPR) bodies, which are implicated in adenovirus genome packaging [78,103,176] (see also the following section on virus-induced assemblies proximal to replication compartments). This raises questions as to how changes in VRCs might functionally impact late viral processes such as viral late gene expression, capsid assembly, and genome packaging. It is also interesting to note that when BK or SV40 DNA is labeled with BrdU, nascent (replicating) DNA localizes to regions that are subtly morphologically distinct from chased (accumulated) DNA, although both nascent and accumulated DNA localize at or in proximity to PML, which is a marker of their respective VRCs [146]. This might suggest that polyomavirus DNA accumulates at a site distinct from where it is replicated, either within the VRC or possibly at distinct nuclear sub-domains proximal to the VRC. The existence of subtly distinct sub-domains within the VRCs of MPyVs have also been suggested [55,153]. With modern advances in the labeling of biomolecules, and super-resolution microscopy techniques, it is likely that much can still be learned about VRC structure and the spatial and temporal organization of the viral processes they harbor. 

## 13. Virus-Induced Assemblies Proximal to Replication Compartments

In addition to VRCs, DNA virus infection can also induce the formation of other nuclear sub-domains, which are often in proximity to VRCs (Figure 2). This raises the possibility that their function and those of VRCs may be linked. However, the functions of these other induced sub-domains are typically less characterized. Infection with HSV-1 induces the formation of virus-induced chaperone enriched (VICE) domains that can be visualized as distinct foci. VICE domains can be observed in a proportion of (but not all) HSV-1 infected cells, are dependent on the viral proteins ICP22 and ICP27, and form juxtaposed to VRCs [177,178,179,180,181]. Heat shock proteins, as well as other components of the cellular protein quality control machinery including ubiquitin and proteasome sub-units, localize to VICE domains [179,180]. Furthermore, domains enriched for newly synthesized viral proteins termed new protein domains (NPDs) can be found juxtaposed or overlapping with VICE domains [182]. These findings support a role for VICE domains in the quality control and degradation of misfolded viral proteins, perhaps prior to recruitment into the VRC. However, more direct evidence of VICE domain function is lacking. Similarly to the formation of VICE domains during HSV-1 infection, assemblies containing components of the protein quality control machinery form at the periphery of VRCs during infection with CMV, raising the possibility that the formation of protein quality control domains may be induced by other herpesviruses and perhaps even other DNA viruses [183]. SSRP1, a component of the cellular facilitates chromatin transcription (FACT) complex, also localizes to foci in a proportion of HSV-1-infected cells, as well as localizing to VRCs. Similarly to VICE domains, SSRP1 foci are also dependent on ICP22, as is the localization of FACT complex components to VRCs [149]. FACT complex components are reported to facilitate HSV-1 gene expression late in infection [64,119,149]. However, the requirement of SSRP1 to form foci and the function of these foci in viral processes is unclear. Furthermore, the relationship of SSRP1 foci to existing or other virus-induced sub-domains such as VICE domains remains to be determined. 

During adenovirus infection, the re-organization of VRCs during late stages of infection coincides with the formation of ViPR bodies. These nuclear assemblies contain accumulated viral DNA, are surrounded by the re-organized VRCs, and contain the nucleolar proteins Myb-binding protein 1 (Mybbp1A), nucleophosmin (NPM1), UBTF, and nucleolin, which have proposed roles in the modification of viral chromatin [14]. It has been suggested that ViPR bodies may function directly in the packaging of viral genomes into capsids, or indirectly in the preparation of genomes for packaging. The adenovirus minor capsid protein IX can also be visualized at virus-induced foci proximal to VRCs [184]. However, the functions fulfilled by these foci are unknown. 

Just as VRCs are essential for the organization of viral processes such as DNA replication and transcription, so too might other virus-induced structures play critical roles in viral processes. Indeed, their functions may be intimately linked to those of VRCs. Thus, further investigation into the function of virus-induced assemblies proximal to VRCs may be highly pertinent to understanding viral processes and how they are organized and coordinated. 

## 14. Conclusions and Future Perspectives

VRCs play an essential and central role in the propagation of nuclear-replicating DNA viruses. The study of these membraneless compartments has driven our understanding of viral processes and the virus–host interactions that regulate them. This highlights factors recruited to VRCs to aid in replication, as well as factors excluded from VRCs to antagonize the restriction of viruses by host processes. However, many interesting questions remain. How do biophysical processes that govern nuclear organization influence the formation and organization of VRCs? How might the microscopic and nanoscopic organization of these compartments facilitate and co-ordinate viral processes? How do the morphological changes in VRCs and the formation of other proximal virus-induced assemblies influence and interplay with VRC function? Understanding the spatial organization of viral genome processes at VRCs at high resolution, and how this organization changes as infection progresses, may be key to understanding how different viral processes are effectively coordinated. This question is particularly interesting when considering the late stage of infection, when the transcription, replication, and packaging/encapsidation of viral genomes are all ongoing. Furthermore, understanding how the existing nuclear architecture must be modified and how biophysical processes can be utilized to drive VRC formation may yet inform our understanding of fundamental nuclear processes. Undoubtedly, as we continue to investigate VRCs and the processes they organize, we stand to learn much about important aspects of both DNA virus and cell biology.

## Figures and Tables

**Figure 1 viruses-12-00151-f001:**
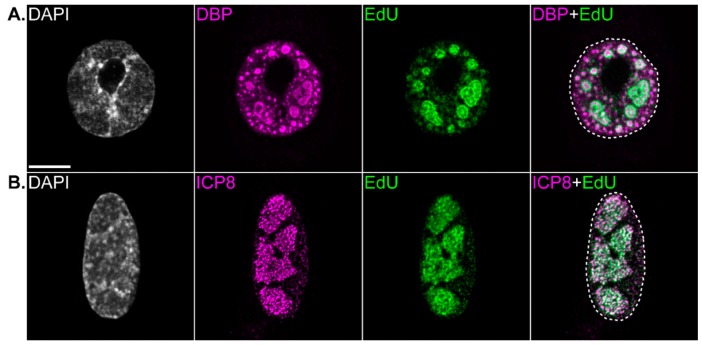
Viral replication compartments are the sites of viral genome replication. Immunofluorescence confocal microscope images showing viral replication compartments in a human adenovirus infected bronchial epithelial cell (**A**), or herpes simplex virus 1-infected foreskin fibroblast (**B**). The nucleus is visualized by the staining of DNA with DAPI. Viral replication compartments are visualized by the immunostaining of viral single-stranded DNA (ssDNA)-binding proteins DNA binding protein (DBP) or infected-cell protein 8 (ICP8), respectively. The sites of viral DNA replication are visualized by incorporating 5-Ethynyl-2’-deoxyuridine (EdU) into DNA during replication (15 min pulse, 10 µM), and subsequently conjugating fluorophores to EdU in a copper-catalyzed cycloaddition reaction. The outline of the nucleus is shown (dotted line). Scale bar = 10 µm.

**Figure 2 viruses-12-00151-f002:**
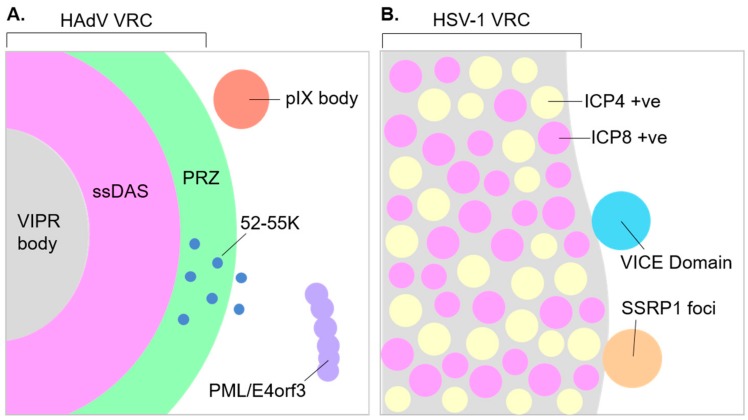
Schematic of human adenovirus and herpes simplex virus 1 replication compartments showing elements of compartment sub-structure and proximal virus-induced domains. (**A**): Partial cross-section of a human adenovirus (HAdV) replication compartment showing virus-induced post-replicative (VIPR) body, single-stranded DNA accumulation site (ssDAS), and peripheral-replicative zone (PRZ). The presence of the viral assembly/packaging protein 52-55K within and proximal to the PRZ is also shown, as is a pIX body, and promyelocytic leukemia protein (PML) track containing the viral early protein E4orf3. (**B**): Partial cross-section of a herpes simplex virus 1 (HSV-1) replication compartment showing infected cell protein 4 (ICP4) and infected cell protein 8 (ICP8) positive regions within, which are associated with transcription and genome replication, respectively. A virus-induced chaperone enriched (VICE) domain and SSRP1 focus are shown at the outer edge of the replication compartment.

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
