# Peer review of "Replication Compartments of DNA Viruses in the Nucleus: Location, Location, Location"

_viruses, 2020, doi:10.3390/v12020151_

Round 1

Reviewer 1 Report

In this manuscript, the authors comprehensively described the significance of virus replication compartment formation for nuclear-replicating DNA viruses. The manuscript was mostly well-written and easy to follow. I have only some minor comments to improve the manuscript.

Title was not fit to this manuscript. It would be better to include ‘nuclear-replicating DNA viruses’ or similar words as the manuscript focuses on nuclear-replicating viruses such as herpesviruses, HAdV, papilloma viruses etc.

In addition, the last ‘location’ should be ‘Location’?

The numbering of section and sub-section is required.

Although this is not essential, I am wondering whether it is possible to expand the section ‘Biophysical processes in replication compartment formation’ (lines 216-237). As the authors mentioned in the text, LLPS is recently a very hot topic in the formation of membrane-less organelles. The manuscript may attract broader readers if includes specific examples of viral or cellular proteins having the ability of LLPS.

Finally, Figure 2 includes only Adenovirus and herpesvirus. Is it possible to include the model of other nuclear-replicating DNA viruses such as SV40, HPVs, or another well studied viruses?

Reviewer 2 Report

This is an excellent review of virus replication, with particular reference to nuclear events for DNA viruses. These include mainly herpes, adeno, parvo, polyoma and papillomaviruses.  We often tend to think of the nucleus in virus-infected cells as a single compartment but it is obviously more complex than this.  With various events taking place at discrete loci within the nucleus, we have much to learn about how this complex organelle facilitates virus replication.

For the virologist who does not specialise in nuclear events, this review offers much in the way of general education to an expert level.  It will be a valuable addition to the literature for both students and established academics alike.

There are only minor points needing correction:

Line 97: It is not clear what "RING" refers to.

Line 143 (and other points throughout): Don't end a page with a heading.

Line 157: "initiated" not "initiation".

Line 391: lower case for "Importance".

Line 484: The sentence beginning, "Highlighting" is a run on from the previous one.  It could begin, "This highlights...".
